# Oxidative Stress and Immune Response in Melanoma: Ion Channels as Targets of Therapy

**DOI:** 10.3390/ijms24010887

**Published:** 2023-01-03

**Authors:** Alessia Remigante, Sara Spinelli, Angela Marino, Michael Pusch, Rossana Morabito, Silvia Dossena

**Affiliations:** 1Department of Chemical, Biological, Pharmaceutical and Environmental Sciences, University of Messina, 98122 Messina, Italy; 2Biophysics Institute, National Research Council, 16149 Genova, Italy; 3Institute of Pharmacology and Toxicology, Paracelsus Medical University, 5020 Salzburg, Austria

**Keywords:** oxidative stress, immune response, cancer, melanoma, ion channels, Ca^2+^ signaling

## Abstract

Oxidative stress and immune response play an important role in the development of several cancers, including melanoma. Ion channels are aberrantly expressed in tumour cells and regulate neoplastic transformation, malignant progression, and resistance to therapy. Ion channels are localized in the plasma membrane or other cellular membranes and are targets of oxidative stress, which is particularly elevated in melanoma. At the same time, ion channels are crucial for normal and cancer cell physiology and are subject to multiple layers of regulation, and therefore represent promising targets for therapeutic intervention. In this review, we analyzed the effects of oxidative stress on ion channels on a molecular and cellular level and in the context of melanoma progression and immune evasion. The possible role of ion channels as targets of alternative therapeutic strategies in melanoma was discussed.

## 1. Introduction

### 1.1. Physiological and Supra-Physiological Oxidative Stress

Oxidative stress is defined as an imbalance between the production of free radicals and/or reactive metabolites, called reactive oxygen and/or nitrogen species, and their elimination by protective mechanisms and chemicals, which are referred to as antioxidants [1]. Some subcellular compartments, such as the endoplasmic reticulum, lysosomes, or peroxisomes, are more oxidizing, whereas others, such as the mitochondria, are more reducing. In fact, the latter are considered the most redox-active compartment in the cell, as they are responsible for more than 90% of oxygen utilization. Although most of the oxygen undergoes complete reduction to water at the level of cytochrome oxidase, partial reduction associated with reactive oxygen species generation could also occur. Then, the levels of reactive species may fluctuate between sub-cellular compartments and may lead to beneficial or pathological effects [2]. In physiological conditions, reactive species may function as second messengers or redox messengers. Cells can generate reactive species and use them for intracellular signaling and stimulating redox-sensitive signaling pathways, or alternatively, to modify the cellular content of the cytoprotective regulatory proteins. To name just a few examples, reactive species may control pro-fibrotic signaling, pro-inflammatory signaling, cell proliferation, apoptosis, and other biological processes without triggering macromolecular damage [3].

Conversely, supra-physiological levels of reactive species may lead to extensive and irreparable cell injury, ultimately resulting in cell death through apoptosis or necrosis. Of all the compounds derived from oxidative metabolism, reactive oxygen species are the most abundantly produced. Their half-lives range from a few nanoseconds to hours, depending on the stability of the chemical species. Reactive oxygen species include hydrogen peroxide (H_2_O_2_), superoxide anion (O_2_^−•^), hydroxyl radical (OH^−^), singlet oxygen (^1^O_2_) and ozone (O_3_) [4,5]. The most abundant reactive nitrogen species is nitric oxide (NO), which is able to react with certain reactive oxygen species, including the superoxide anion (O_2_^−^), to produce peroxynitrite (ONOO^−^). Nitric oxide and peroxynitrite can also be converted into peroxynitrous acid (HNO_3_) and ultimately into hydroxyl radical (OH^−^) and nitrite anion (NO_2_^−^) [4,6,7,8].

The imbalance between oxidants and antioxidants leads to damage of cellular biomolecules, with potential impacts on the whole organism [9,10,11,12]. Specifically, reactive species cause nicks in the DNA and malfunction in the DNA repair mechanisms. DNA oxidation by these reactive species generates 8-hydroxy-2′-deoxyguanosine, a product that is able to generate mutations in DNA during carcinogenesis. Lipids and proteins are also significant targets of oxidative attack, and modification of these molecules can increase the risk of mutagenesis [9]. The cellular plasma membrane is rich in polyunsaturated lipids that are susceptible to oxidation by reactive species [13]. The reactive species initiate lipid peroxidation reactions and consequently increase the permeability of the cell membrane, which could lead to cell death. For example, by inducing excessive lipid peroxidation, reactive nitrogen species can further generate other reactive species, such as reactive aldehydes-malondialdehyde (MDA) and 4-hydroxynonenal (4-HNE). Cellular proteins are the most affected by high concentrations of reactive species. Proteins suffer from the generation and accumulation of carbonyl groups, such as aldehydes and ketones, or alternatively, from the conversion of thiol groups (–SH) into sulfur reactive radicals. Due to these oxidation-induced modifications, alterations in the protein structure may occur and thereby loss and/or changes in protein function may result [14].

### 1.2. Endogenous and Exogenous Antioxidants

The redox balance in the cell is normally regulated by a complex antioxidant system. Antioxidants can be variably classified into different groups according to their chemical properties and origin. Endogenous antioxidants include glutathione, alpha-lipoic acid, coenzyme Q, melatonin, catalase (CAT), superoxide dismutase (SOD), thioredoxins (TRX), glutathione peroxidases (GPXs) and peroxiredoxins (PRXs). Natural antioxidant compounds can be obtained from the diet, e.g., beta-carotene (vitamin A), alpha-ascorbic acid (vitamin C), tocopherol (vitamin E) and polyphenol metabolites. Examples of synthetic antioxidants include pyruvate, selenium, N-acetyl cysteine (NAC), butylated hydroxytoluene, butylated hydroxy-anisole, and propyl-gallate [15,16].

### 1.3. Oxidative Stress in Human Disease

Oxidative stress is important from a biomedical point of view because it is related to a wide variety of human diseases [17], including neurodegenerative disease [18,19], cardiovascular disease [20], inflammatory disease [21], diabetes [22], aging [23] and cancer [24]. Many investigations have demonstrated a direct relationship between chronic oxidative stress and carcinogenesis [3]. In cancer cells, high levels of reactive oxygen species and/or reactive nitrogen species can result from increased metabolic activity, mitochondrial dysfunction, peroxisome activity, increased cellular receptor signaling, oncogene activity, increased activity of oxidases, cyclooxygenases, lipoxygenases and thymidine phosphorylase, or through crosstalk with infiltrating immune cells [25,26]. Reactive species-sensitive signaling pathways are persistently elevated in many types of cancers, including melanoma, where they participate in cell growth/proliferation, differentiation, protein synthesis, glucose metabolism, cell survival, and inflammation [27,28,29,30].

### 1.4. Melanoma Pathophysiology and Current Options for Treatment

Melanoma, a neoplasm arising from malignant transformation of melanocytes, is the most lethal form of skin cancer [29]. However, melanoma can also develop on mucosal surfaces such as the oral cavity, the genital mucosa, the upper gastrointestinal mucosa as well as the uveal tract of the eye and leptomeninges [31]. The incidence of cutaneous melanoma has rapidly increased over the past decades. Melanoma is the ninth most common malignancy and the second for mortality, with an incidence being markedly increased in patients with a history of heavy sun exposure or isolated episodes of serious sunburn [32,33]. Although the majority of primary melanomas are cured with local wide excision, metastatic melanoma carries a grim prognosis, with a median survival of nine months and a long-term survival rate of 10% [34]. Cancer metastasis is considered the end stage of the progression of any tumour. It is composed of different steps that include infiltration of cancerous cells into the neighboring tissue, followed by intravasation as tumour cells undergo trans-endothelial migration through the vessel wall and, finally, extravasation and proliferation at the distant organ to form secondary tumours [35,36]. About half of all melanomas carry mutations in the BRAF gene, which makes these tumours amenable to targeted therapy. BRAF V600 mutation-positive unresectable or metastatic melanoma in adults is treated with the selective competitive inhibitor of BRAF kinase dabrafenib as monotherapy or in combination with the MEK inhibitor trametinib. Other treatment options are represented by the immune checkpoint inhibitors, which include the PD-1 inhibitors nivolumab and pembrolizumab and the CTLA-4 inhibitor ipilimumab. Second-line therapy is achieved by chemotherapy with alkylating cytostatic dacarbazine (DTIC), amongst others. Radiation therapy can be a useful treatment in some clinical settings including adjuvant therapy after complete excision of a primary melanoma or after therapeutic lymphadenectomy [37,38].

### 1.5. Oxidative Stress and Melanocytes

A distinctive feature of melanoma compared to other solid tumours is the especially high oxidative stress level, which can be explained by both extrinsic and intrinsic factors [31,39,40]. Due to their physical location, melanocytes are directly exposed to environmental factors inducing oxidative stress, such as UV radiation [41,42]. Epidemiological studies have demonstrated a strong association between UV radiation and melanoma risk. UV light is a type of electromagnetic radiation emitted by the sun. The UV spectrum is conventionally subdivided into UVA radiation (320–400 nm), UVB radiation (280–320 nm), and UVC radiation (100–280 nm). Only UVA radiation and a portion of the UVB spectrum (above approximately 300 nm) can reach the surface of the earth. Thus, both UVA and UVB may contribute to melanoma development [43]. UV radiation can lead to indirect oxidation-mediated damage of cutaneous macromolecules by stimulating reactive oxygen species production through enzymatic reactions catalyzed by enzymes such as NADPH oxidase (NOX1 and NOX4), cyclo-oxygenase, and xanthine oxidase, or by the damage of mitochondrial respiratory chain enzymes. Alternatively, UV induces the skin to also produce high levels of reactive nitrogen species (NO and possibly ONOO^−^) [44]. When UV-stimulated reactive oxygen species target DNA molecules, various types of oxidative DNA lesions are induced, including DNA single-strand breaks, DNA–protein crosslinks, and alteration of DNA nitrogenous bases. In particular, the oxidation of the guanine bases, which produces 8-oxo-7,8-dihydroguanine (8-oxoG), is the most abundant form of oxidative DNA damage [45] (Figure 1). These alterations can induce inflammation and can further initiate tumorigenesis [46]. Reactive species and damaged DNA can activate intracellular protein complexes such as inflammasomes [30,47]. In this context, both keratinocytes and melanocytes secrete cytokines with pro-inflammatory action, thus modulating innate and adaptive immune responses [48]. All the immune-related molecules, cytokines, chemokines, and non-immune molecules, such as growth factors have both paracrine and autocrine effects upon the microenvironment and design the local milieu that initiates and then regulates local inflammation or can lose control, consequently favouring the process of tumorigenesis. Inflammation has acute and chronic stages, but its link to tumorigenesis is carried out by chronic inflammation [49]. During inflammatory response, mast cells and monocytes/macrophages are recruited [50]. In particular, mast cells are the first to migrate to the site of proliferation; macrophages follow later in the response. Both are capable of producing reactive species as a cytotoxic mediator to kill cells [51]. Reactive oxygen species can react with the nucleic acids attacking the nitrogenous bases and the sugar phosphate backbone and can evoke single- and double-stranded DNA breaks [52]. While acute inflammation is regulated by T-helper (Th)1-polarized T lymphocytes attracted by innate immune cells, secreting mainly anti-tumour immune molecules such as interleukin (IL)-2 and interferon (IFN)-γ, chronic inflammation is controlled by regulatory T cells (Tregs), Th2 cells, that secrete pro-tumorigenic factors (IL-4, IL-6, IL-10, IL-13 and transforming growth factor (TGF)-β) [53]. In this regard, reactive species produced by melanoma cells and tumor-infiltrating leukocytes, including Tregs, can suppress immune responses [52].

Additionally, UV-induced reactive oxygen species also attack other major biomolecules, causing protein oxidation and lipoperoxidation that compromise cellular ultrastructure and function [54]. In fact, when UV radiation hits the skin, within sebaceous lipids, squalene is oxidized and can initiate inflammatory processes, thus acting as an inflammasome activation danger signal (Figure 1). In this regard, it is worth mentioning that melanocytes are more vulnerable to UV-mediated oxidative injury than other skin cells, such as keratinocytes and fibroblasts, since their specialized function, namely the melanin synthesis, is an energy-consuming process that itself contributes to generating a large amount of reactive oxygen species [40]. In fact, there are conflicting data in the literature on the pro-oxidant and antioxidant effects exerted by melanin [55]. The presence of melanin in the skin appears to be a double-edged sword: it protects melanocytes through its capacity to absorb UV radiation, but its synthesis in melanocytes results in higher levels of intracellular reactive oxygen species that may increase melanoma susceptibility. During the melanin biosynthesis, tyrosinase enzyme oxidizes tyrosine to L-DOPA, which itself is oxidized to DOPA-quinone, a reactive molecule toward thiols and/or amino groups. Afterward, a redox exchange converts the DOPA-quinone into DOPA-chrome, which, after a decarboxylation yields dihydroxy-indole, or alternatively, after tautomerization produces dihydroxy-indole carboxylic acid. The process that converts indoles to quinones implicates an important generation of reactive oxygen species (O_2_^•−^ and H_2_O_2_) (Figure 1) [56,57]. Finally, the polymerization of the quinones results in the formation of black-brown eumelanin. Instead, the pheomelanin, which displays a typical red-yellow colour, differs from the eumelanin for having a higher ratio of sulphur to quinones, and its biogenesis process has as intermediate the generation of cysteinyl-DOPA instead of L-DOPA. These variations are responsible for the higher pro-oxidant effects caused by the sunlight of pheomelanin with respect to eumelanin. Eumelanin is a good free radical scavenger; pheomelanin is not, and its benzothiazole units can act as photosensitizers leading to the production of reactive oxygen species [53,57,58,59,60]. Paradoxically, while high levels of reactive oxygen species can cause oxidative stress and induce cell death, low levels of superoxide and H_2_O_2_ can promote G1→S cell cycle transition. Thus, oxidative stress or redox status shifts may cause cell transition from a quiescent to a proliferative status, growth arrest, or cell death, according to the duration and extent of the redox imbalance [61].

**Figure 1 ijms-24-00887-f001:**
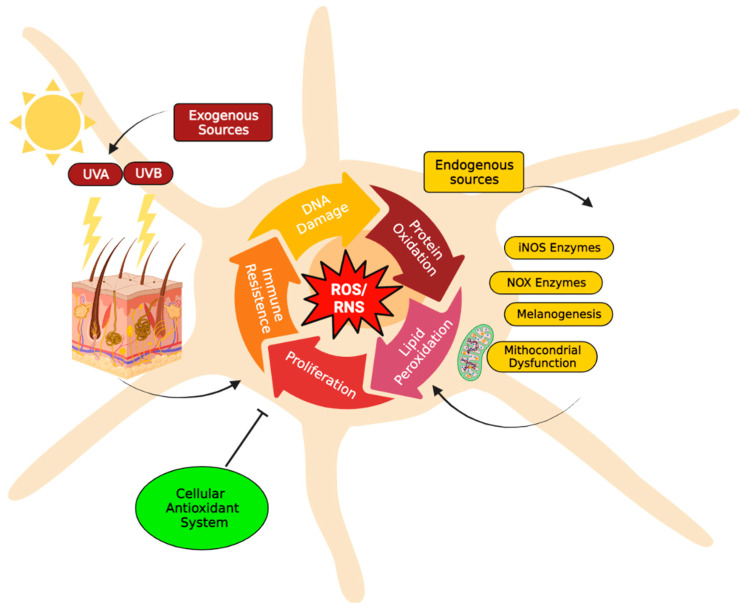
Major reactive species sources in melanocytes. The increase of reactive oxygen species (H_2_O_2_, O_2_^−•^) and/or reactive nitrogenous species (NO and ONOO^−^) induces severe damages to major biomolecules, resulting in DNA and protein oxidation, as well as lipoperoxidation, that compromise cellular structure and function. Consequently, these alterations can induce inflammation and can initiate tumorigenesis processes (e.g., cell proliferation and adaptive immune resistance). To maintain acceptable levels of reactive species, melanocytes cells usually increase their antioxidant systems to protect cells from oxidative stress damage and restore physiological redox balance. The redox balance in the cell is normally regulated by a complex antioxidant system. Endogenous antioxidants include catalase (CAT), superoxide dismutase (SOD), glutathione peroxidases (GPXs) and glutathione (GSH). In particular, GSH metabolism protects melanocytes from the toxic effects of H_2_O_2_ formed during melanin synthesis. GSH metabolism, therefore, appears to be critically important to the maintenance of melanocyte cell viability [62]. Instead, natural antioxidant compounds can be obtained from the diet, e.g., beta-carotene (vitamin A), alpha-ascorbic acid (vitamin C), tocopherol (vitamin E) [44]. The figure was created using BioRender.com.

### 1.6. Melanocytes and Immune Response

Accumulating evidence supports the concept that melanocytes are not only professional melanin-producing cells but are also active factors in the cutaneous immune system [63,64]. The production of melanin involves stepwise oxidation of the amino acid tyrosine and downstream aromatic compounds. Myelinization has important protective roles in several species, as toxic intermediates (semi-quinone, DOPA-quinone and indole-quinone) may be produced, including reactive oxygen species. These intermediate compounds are believed to exert strong antimicrobial activities, and melanin, the end-product of myelinization, may have the capacity to trap, inhibit, and even kill invading bacteria and other microorganisms [65,66].

Melanin may also have a crucial immune-regulatory role. It has been found to have immune-modulatory activities through inhibition of pro-inflammatory cytokine production by T lymphocytes, monocytes, fibroblasts, and endothelial cells. The transfer of acidified melanin-containing organelles (melanosomes) from melanocytes to neighboring keratinocytes in the outer portions of the epidermis may have a role in acidifying the stratum corneum in darkly pigmented skin [67]. Acidity in the stratum corneum could enhance skin barrier function and the integrity and/or cohesion of stratum corneum; it might also exert antimicrobial function [68]. In response to different stimuli, melanocytes could also regulate cutaneous immune response by producing and releasing several immune-suppressive molecules, e.g., alpha-melanocyte stimulating hormone (a-MSH). The latter participates in both anti-inflammatory and immunomodulatory activities [69]. In this context, it has also been demonstrated that melanocytes are capable of phagocytosis. In this regard, melanosomes have functional and structural similarities to lysosomes, and have been considered as indeed specialized lysosomes. Because phagocytosis is understood to be a prerequisite for antigen processing and presentation, phagocytosis by melanocytes suggests that the melanocytes have antigen presentation potential [63]. Finally, human melanocytes express functional toll-like receptors (TLRs). Upon ligation of TLRs with lipopolysaccharide, these cells may trigger NF-kB and/or mitogen-activated protein kinase signaling pathways, thus producing several pro-inflammatory cytokines and chemokines. These molecules may modulate the recruitment and activation of different immune cells in the skin. Thus, the expression of functional TLRs on melanocytes suggests that they may act as early sensors in immune responsiveness [70].

### 1.7. Immune Evasion in Melanoma and Potential Novel Options for treatment

Melanoma is also an immunologic malignancy [71]. These cancer cells are constantly adapting to the host defenses by manipulating intrinsic and extrinsic biological pathways [71]. In the event of the onset and development of melanoma, the immune system is exposed to numerous previously unseen antigens that are derived from genetic abnormalities. In this context, the immune system can recognize and eliminate some cancers at an early stage of their development. The adaptive immune system appears to be of fundamental importance in the antitumor response, which is triggered by activation of a wide range of diverse and highly specific receptors on T and B cells. An effective immune response begins when the T or B cells recognize the tumor antigen in a pro-stimulatory context and undergo activation and proliferation. B cells have as a receptor a surface IgM immunoglobulin and are able to recognize soluble antigens, bind to them and differentiate into plasmacytes, which secrete large amounts of highly specific antibodies [72]. Unfortunately, melanoma cells may develop numerous immuno-evasive mechanisms that allow them to resist natural or therapy-induced immune attacks [73]. Through these mechanisms, tumour cells are capable of modulating themselves and their surroundings in order to promote their survival, growth, and invasion, even under persistent immune pressure. Indeed, stressors present in the tumour microenvironment, such as chronic hypoxia, play crucial roles in promoting tumour cell plasticity and heterogeneity, which finally leads to the acquisition of immune tolerance and tumour progression [74,75]. The plasticity of melanoma cells leads to a phenomenon called immune escape, whereby cancer cells acquire a less immunogenic phenotype and the ability to suppress anti-tumour immune cells within the tumour microenvironment [49,76,77]. Although the introduction of the immune checkpoint inhibitors mentioned above has undoubtedly represented a great advancement in the treatment of melanoma and has improved patient prognosis, many patients do not respond to therapy and consequently remain with limited options for treatment. Novel treatment options might include newer checkpoint inhibitors such as B- and T-lymphocyte attenuator (BTLA), lymphocyte- activation gene 3 (LAG-3), and T-cell immunoglobulin and mucin domain-3 (TIM-3) inhibitors [78]. These are subject of intense investigation in preclinical and clinical studies [79,80,81].

A possible alternative strategy to improve therapeutic efficacy can be targeting the redox balance in cancer cells [82]. Ion channels are transmembrane proteins that connect the inside of the cell to its outside in a selective fashion by regulating the ionic permeability of cell membranes. Ion channels represent an important class of biomolecules due to their ability to serve as key elements in signaling and sensing pathways [83,84,85,86]. Over the past 10 years, it became obvious that ion channels play a key role in cancer development by influencing cell migration, cell cycle progression, and proliferation [87,88,89,90,91]. During the transition from a normal cell towards a cancer cell, a series of genetic alterations occur, which may also affect ion channel expression, or may cause a change in ion channel activity. To name just some examples, cell migration is important not only for initiation of metastasis [92], but also plays a critical role for the homing of tumour-infiltrating lymphocytes [93]. In addition, ionic (calcium) signalling might influence the tumour microenvironment and change the fate of the melanoma by altering the function of innate and adaptive immune cells and regulating extracellular matrix and tumour vascularization, thus adapting to different physical and chemical surroundings [94]. Since ion channels are mostly localized to the plasma membrane, they can be subjected to multiple layers of regulation, and therefore represent promising targets for therapeutic intervention in cancer. Indeed, reactive oxygen species production can directly induce post-translational modification of ion channels leading to oxidation and/or nitration of specific amino acid residues or indirectly modulate channel function by affecting the intracellular signaling pathways [95].

In this mini-review, the case is made that ion channels are critically important for tumour growth and metastasis and, therefore, are potential targets in the pharmacological treatment of melanoma. Furthermore, the role of ion channels in the redox signaling between melanoma cells and immune cells is described.

## 2. Calcium Channels

Calcium is a second messenger ion that acts as a regulator of important cellular functions such as migration, proliferation, and differentiation of tumour cells, including melanoma cells [94]. Indeed, several studies have shown the contribution of Ca^2+^ homeostasis to well-known oncogenic signalling pathways [96]. Calcium channels are widely expressed on several biological membranes, such as the mitochondrial, endoplasmic reticulum, and plasma membranes. The calcium entry channels can be divided into receptor-operated calcium channels (ROCCs), voltage-dependent calcium channels (VDCCs), and store-operated calcium entry (SOCE) on the plasma membrane [97]. In particular, store-operated Ca^2+^ entry (SOCE) is a mechanism by which Ca^2+^ release from the endoplasmic reticulum stimulates Ca^2+^ entry from the extracellular space [98,99]. This influx is mainly operated by ORAI channels located in the plasma membrane and the endoplasmic reticulum stromal interaction molecules (STIMs) [100,101]. Growing evidence indicates that a robust SOCE response is responsible for the stimulation of proliferation, survival, and cell migration of cancerous cells, including melanoma cells [100,102,103]. Elevated levels of reactive oxygen species are a typical feature of the microenvironment in melanoma. A recent publication by Gibhardt and collaborators addressed the regulatory mechanisms by which STIM2 is regulated by oxidative stress in melanoma cells. Specifically, these authors identified two cysteines in the STIM2 protein, C302 and C313, which can be modified following exposure to H_2_O_2_. Functional studies revealed that oxidative stress-induced C313 sulfonylation hindered STIM2 oligomerization, thus causing inhibition of the association between STIM2 and ORAI1 and thereby SOCE [104,105]. This study is an interesting demonstration of the sensor–transducer function of an ion channel-associated protein. In fact, STIM2, together with ORAI1, promotes growth and invasion of melanoma cells [106]. These results suggest that inhibition SOCE mediated by cysteine oxidation of STIM2 could thereby block migration and invasion of this highly aggressive cancer (Figure 2). Then, the intracellular calcium signaling, and the multiple channels involved in its control, function as regulators of melanoma progression and could serve as mechanistic targets for the suppression of melanoma growth and management of metastasis.

The melanogenic process is very complex. There is a crosstalk of melanoma cells with inflammatory immune cells, which significantly influences the biology of melanoma in terms of proliferation, differentiation, and progression [78]. The inflammatory response induces the recruitment of innate and adaptive immune cells that could indirectly cause oxidative stress [107]. In this regard, it must be considered that the redox regulation of STIM2 also occurs in lymphocytes. This raises the question as to whether this mechanism could contribute to the immuno-suppressive properties of the tumour microenvironment and thereby might contribute towards explaining the lack of therapeutic success of conventional chemotherapeutics in melanoma [105].

Another class of Ca^2+^ channels implicated in melanoma is the family of transient receptor potential (TRP) channels [103,108]. Specifically, transient receptor potential melastatin 2 (TRPM2) is a 1503-amino acid channel implicated in several pathological pathways involving oxidative stress and its activation has been generally associated with a large increase in intracellular Ca^2+^ levels [108]. Among oxidant agents, chloramine-T (Chl-T) efficiently oxidizes methionines and represents an ideal investigational tool to mimic abnormal intracellular oxidative stress levels. In this regard, Ferrera and co-authors demonstrated that Chl-T activated TRPM2 channel in a melanoma cell line (IGR39), thus inducing a dramatic increase in intracellular Ca^2+^ content [109]. In cell culture, higher expression of TRPM2 induced melanoma cell proliferation, migratory ability, and invasiveness [110] (Figure 2). Moreover, functional activation of TRPM2 has been associated with highly metastatic melanoma cells [94]. The sensitivity of TRPM2 to oxidation is particularly intriguing and might be important in the tumour microenvironment, which is characterized by increased oxidative stress [111]. Thus, pharmacological targeting of TRPM2 might represent a novel therapeutic approach to reduce the metastatic potential of melanoma cells and could complement classical radiation therapy or chemotherapy.

Transient receptor potential ankyrin 1 (TRPA1), belonging to a subgroup of TRP channels, also appears to contribute to melanoma progression. De Logu and collaborators demonstrated that H_2_O_2_ evoked a TRPA1-dependent Ca^2+^ response in two distinct melanoma cell lines (SK-MEL-28 and WM266-4), thus promoting anti-apoptotic and pro-oncogenic programs (Figure 2). This Ca^2+^ influx was prevented by the TRPA1 antagonist A967079. Importantly, H_2_O_2_ elicited a TRPA1-dependent H_2_O_2_ release attenuated by TRPA1 pharmacological antagonism or gene silencing.

## 3. Potassium Channels

Big Potassium (BK) voltage-and Ca^2+^-activated ion channels allow potassium ions to pass selectively, and their gating is subject to physio-pathological and pharmacological regulation. Consequently, these channels became candidate targets of cancer therapeutics, although their role is complex and may not be universal. BK channels are over-expressed in some cancers, including melanoma [112,113,114]. Recently, it has been found that reactive oxygen species modulate BK channel activity [115]. In IGR39 cells, a primary melanoma cell line (IGR39), Ferrera and collaborators demonstrated that an increase in oxidative stress following exposure to Chl-T stimulated the activation of two different Ca^2+^-dependent K^+^ channels: the large-conductance voltage-dependent BK channel encoded by *KCNMA1*, and the medium-conductance voltage-independent KCa3.1 channel encoded by *KCNN4*. The activation of these channels was linked to an increased [Ca^2+^]_i_ consequent to activation of TRPM2 (Figure 2). Conversely, in the metastatic cell line IGR37, no potassium currents were activated upon Chl-T application and no increases in intracellular Ca^2+^levels were detected, in line with the lower expression of *KCNMA1* and *KCNN4* genes in these cells [109]. The high expression and activity of BK and KCa3.1 channels in melanoma cells and the concomitant [Ca^2+^]_i_ increase observed in response to oxidative stress support a role of these molecular entities in the progression of melanoma. In general, the increase in intracellular Ca^2+^ levels induced, for example, by TRPM activation during Chl-T application [109], could be accompanied by K^+^ efflux through BK channels, which leads to membrane hyperpolarization and increases the driving force for Ca^2+^ entry across the plasma membrane, thus promoting cancer cell migration. In addition, Ca^2+^ is a crucial messenger in the mitogenic signal cascade of melanoma cells [116]. Thus, the inhibition of BK channel and related Ca^2+^ influx represents a possible tool to specifically inhibit tumour cell proliferation [116]. Additionally, KCa3.1 channels were found over-expressed in Mel Im cells -a melanoma cell line- and were implicated in the promotion of cell migration. Cell migration is a very complex phenomenon. For adherent cells to move in a directional fashion, multiple coordinated processes need to occur, including extension of lamellipodia at the front of the cell, disassembly of focal adhesions at the rear of the migrating cell, and force generation through cytoskeleton attachments to the extracellular matrix to pull the cell forward [117]. In migrating cells, KCa3.1 channel activity was detected predominantly at the rear cell pole, which may be due to the intracellular Ca^2+^ gradient in polarized, migrating cells. Interestingly, Schmidt and collaborators found that KCa3.1 supports secretion of melanoma inhibitory activity (MIA) protein at the cell rear. MIA protein is known to play a key role in melanoma development, progression, and formation of metastasis. This secretion was diminished by the specific KCa3.1 channel inhibitor TRAM-34 and KCa3.1 channel mutation. These findings disclosed a new mechanism by which KCa3.1 potassium channels promote cell migration [118] (Figure 2).

Another potassium channel implicated in melanoma is TASK-3, which belongs to the superfamily of the twin-pore domain potassium channels. These channels are thought to promote proliferation and/or survival of malignant cells, most likely by increasing their tolerance to hypoxia [119]. The tumour-promoting effect of TASK-3 channels may not be related to potassium movement through the plasma membrane but seems rather linked to the involvement of these channels in maintaining mitochondrial potassium homeostasis. During oxidative stress increase, the effect of reduced TASK-3 expression on the mitochondrial function and cell survival has been investigated in WM35 and A2058 melanoma cell lines. TASK-3 channels are functionally expressed in the mitochondria, underscoring their contribution to the mitochondrial function, and reduced TASK-3 biosynthesis impeded the mitochondrial activity of melanoma cells. Further, TASK-3 knockdown led to reduced cell viability and increased apoptosis [120]. Therefore, also TASK-3 channels may be considered as targets when designing new apoptosis-inducing therapeutic strategies.

Some of the molecular mechanisms associating ion channels with melanoma include hypoxia, immune response, as well as non-canonical functions of ion channels. This diversity of mechanisms offers an exciting possibility to suggest novel and more effective therapeutic approaches to fight cancer [121]. Alterations in microRNA (miRNA) profiles induced by hypoxia allow cancer cells to acquire immune-resistance phenotypes. miRNAs are 19–22 nucleotide long non-coding RNAs that regulate gene expression at a post-transcriptional level. During the last few years, studies on microRNA (miRNA) and cancer have burst onto the scene. Their profile and levels of miRNAs are frequently altered in cancer cells and affect tumorigenesis and cancer progression [122,123]. In particular, hypoxia-induced miRNAs have been implicated in cancer progression through numerous mechanisms, including the direct transfer of hypoxia-responsive miR-192-5p from melanoma to immune cells–dendritic cells and melanoma-specific cytotoxic T-lymphocytes- via gap junctions [124] (Figure 2). Gap junctions are channels built from proteins called connexins and allow for the exchange of small molecules and K^+^, Na^+^ or Cl^-^ ions with the extracellular environment or adjacent cells [125]. Thus, Connexin 43 (Cx43) might represent a mechanism of immune evasion used by hypoxic melanoma cells [124]. This concept can be exploited for the design of strategies aimed to improve the outcome of immunotherapy for hypoxic cancers.

## 4. Conclusions

Melanocytes are particularly susceptible to oxidative stress owing to the pro-oxidant state generated during synthesis of melanin and the intrinsic antioxidant defenses that may be shattered in pathologic conditions. Oxidative stress can disrupt the homeostasis of melanocytes and cause damage to DNA, proteins, and other cellular components. Altered levels of reactive oxygen species could also affect epigenetic mechanisms and promote alterations in gene expression, thus leading to tumour development and cancer growth. Increases in oxidative stress can directly induce post-translational modification of ion channels or indirectly modulate channel activity by affecting the signaling pathways that control gene transcription as well as protein trafficking and turnover. The activity of ion channels can be stimulated or blocked during oxidative stress, and, in turn, these molecular entities can be involved in determining or attenuating melanoma progression. Therefore, identifying possible alterations in the expression and activity of ion channels in melanoma and the corresponding pathophysiological implications could lead to the identification of alternative treatments, which could be especially useful for patients who do not respond to standard and/or immune therapy or develop resistant tumours.

## Figures and Tables

**Figure 2 ijms-24-00887-f002:**
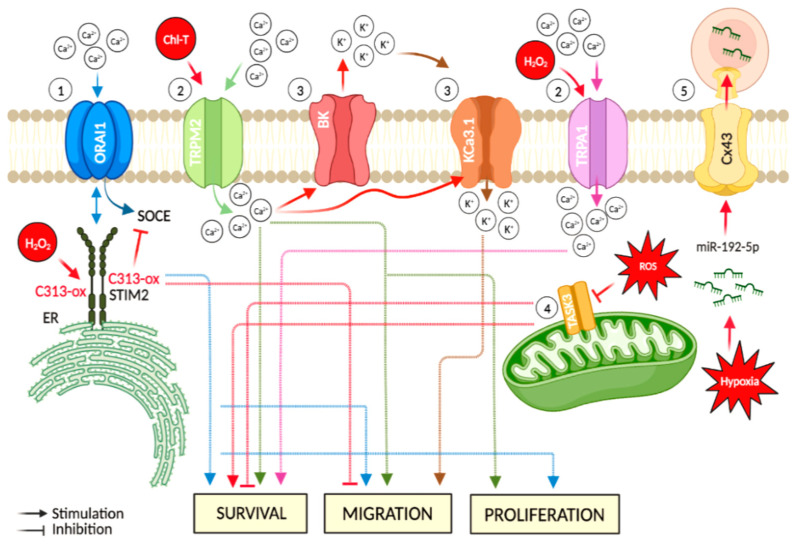
Major hypoxia and cellular oxidative stress-dependent mechanisms involving ion channels in melanoma. Events directly linked to oxidative stress are represented in red: (1) the ER-residing STIM2 protein gates ORAI Ca^2+^ channels at the plasma membrane. This mechanism acts as a regulator in cancer-associated processes, such as cell migration. Oxidative stress-induced C313 sulfonylation hinders STIM2 oligomerization, thus causing inhibition of SOCE; (2) increased oxidative stress leads to up-regulation of TRP channels (e.g., TRPM2 and TRPA1) and induces an increase of intracellular Ca^2+^ levels, thus promoting cancer progression; (3) oxidative stress-induced increase of intracellular Ca^2+^content is accompanied by K^+^ efflux via BK channels, which preserves the ion balance and helps to maintain the Ca^2+^ entry, thus promoting cancer cell migration. Moreover, this oxidation condition stimulates the activation of the medium-conductance voltage-independent KCa3.1, a target implicated in the promotion of cell migration; (4) during oxidative stress, TASK-3 channel down-regulation impedes the mitochondrial activity of cancer cells. In addition, TASK-3 knockdown in melanoma cells leads to reduced viability and increased apoptosis; and (5) hypoxic melanoma cells activate a mechanism of immune evasion through the miR-192-5p efflux by connexin 43 channel (Cx43). The figure was created using BioRender.com.

## Data Availability

Not applicable.

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
