# Peer review of "Oxidative Stress and Immune Response in Melanoma: Ion Channels as Targets of Therapy"

_ijms, 2023, doi:10.3390/ijms24010887_

Round 1

Reviewer 1 Report

Manuscript N.: ijms-2094622Title

Oxidative Stress and Immune Response in Melanoma: Ion 

Channels as Targets of Therapy 

Alessia Remigante, Sara Spinelli, Angela Marino, Michael Pusch, Rossana Morabito and Silvia Dossena 

 Comments:

Remigante and coworkers have made a review with the aim of describing the role of oxidative stress and immune response in the development of melanoma. In particular, the authors examine oxidative stress's molecular and cellular effects on ion channels in relation to the development of melanoma and immune evasion. Finally, they also discussed if ion channels could serve as targets for alternate therapeutic approaches in melanoma cases.

Overall, the authors have made a good a good report with a detailed literature review. 

I only suggest some modifications:

1)        I suggest splitting the “Introduction” into subsections or in new paragraphs to help a reader to follow all the points discussed within it.

2)        Always to help the readability of the Introduction, I think more Figures can be helpful.

Author Response

Remigante and coworkers have made a review with the aim of describing the role of oxidative stress and immune response in the development of melanoma. In particular, the authors examine oxidative stress's molecular and cellular effects on ion channels in relation to the development of melanoma and immune evasion. Finally, they also discussed if ion channels could serve as targets for alternate therapeutic approaches in melanoma cases.

Overall, the authors have made a good a good report with a detailed literature review. We thank the Reviewer for the overall positive evaluation.

I only suggest some modifications:

1) I suggest splitting the “Introduction” into subsections or in new paragraphs to help a reader to follow all the points discussed within it. We thank the reviewer for this suggestion. Done

2) Always to help the readability of the Introduction, I think more Figures can be helpful. We thank the reviewer for this suggestion. Nonetheless, we thought  opportune not to add new Figures to support the introduction and only divided this section into subsections, in an attempt of course to let the reader better follow each discussed point and, actually, to match as much as we can the indications of both Reviewer 1 and Reviewer 2 together,  as Reviewer 2 suggested to summarize the Introduction.  In this light, we considered that more Figures in the Introduction may be inappropriate.

Reviewer 2 Report

This review surveys the role of oxidative stress and immune response in melanoma and attempts to focus on ion channels as targets of therapy. The authors spend too much time in the beginning with a general introduction and towards the end, they have wrapped up the article without providing enough information relating to the cross-talk between melanocytes, immune cells, and the role of oxidative stress.

Major comments:

-          The first two pages of the article contain a brief introduction to the ROS/RNS species. The authors must include a figure to demonstrate the various reactive species generated.

-          The authors must also include a figure to illustrate the reactive species that are generated by endogenous and exogenous sources and the various antioxidant defense mechanisms that restore normal redox state in melanocytes and the interaction with immune cells. What are the ROS/RNS species generated during the melanin synthetic pathway?

-          Although the article title involves oxidative stress and immune response in melanoma, there is very little information provided on the immune response end. How do ROS/RNS species regulate immune cells? How do these reactive species promote melanoma growth and inflammation that results in an immune-suppressive tumor microenvironment? Various studies should be highlighted. For example, studies by Greten FR, Grivennikov SI. Et. al. should be included.

-          On line 204, the authors state that – “These are subject of intense investigation in preclinical and clinical studies (PMID: 33718105; PMID: 35170503; PMID: 32601081).” The authors must remain consistent while citing references and studies. Throughout the article, the authors have used the numbering system, but here they have used the PMID.

Minor comments:

-          On line 414 different font size has been used – “Altered levels of reactive oxygen species could also affect epigenetic mechanisms and promote alterations in gene expression, thus leading to tumor development and cancer growth.”

Reviewer 3 Report

The review article deals with a universally discussed item, i.e., sufficient role of reactive oxygen species (ROS) as a damaging factor of carcinogenesis and tumor evolution. An original line of discussion is traced towards potential role of certain ion channels in maintaining malignant growth exemplified by melanoma. This type of malignancy is highly exposed to exo- and endogenous ROS generated from different sources.  Special attention is drawn to damage of cell membrane lipids and proteins, thus, potentially, leading to impaired functioning of ion channels. In particular, the authors consider calcium transport and probable targeting of potassium channels, in order to amplify the oxidation effects and induce apoptosis of cancer cells. Thus, in view of authors, the manipulation with ion channels could be an alternative way of anticancer therapy. Since most data on the subject are obtained, in vitro, this opinion is rational, though largely hypothetic.

Generally, the review is rather useful, contains original views, does not contradict current knowledge and could be published in the Journal.

There are no gross remarks. One could significantly abridge the volume of data about ROS origin and effects in living cells (pages 2 and 3) – these data are well-known and may be presented in brief form (1/2-1 page) or with a table.

Moreover, only minimal language editing is required.

Author Response

The review article deals with a universally discussed item, i.e., sufficient role of reactive oxygen species (ROS) as a damaging factor of carcinogenesis and tumor evolution. An original line of discussion is traced towards potential role of certain ion channels in maintaining malignant growth exemplified by melanoma. This type of malignancy is highly exposed to exo- and endogenous ROS generated from different sources.  Special attention is drawn to damage of cell membrane lipids and proteins, thus, potentially, leading to impaired functioning of ion channels. In particular, the authors consider calcium transport and probable targeting of potassium channels, in order to amplify the oxidation effects and induce apoptosis of cancer cells. Thus, in view of authors, the manipulation with ion channels could be an alternative way of anticancer therapy. Since most data on the subject are obtained, in vitro, this opinion is rational, though largely hypothetic.

Generally, the review is rather useful, contains original views, does not contradict current knowledge and could be published in the Journal. We thank the Reviewer for the overall positive evaluation.

There are no gross remarks. One could significantly abridge the volume of data about ROS origin and effects in living cells (pages 2 and 3) – these data are well-known and may be presented in brief form (1/2-1 page) or with a table. We thank the reviewer for this suggestion. As suggested by Reviewer 1, the Introduction was divided into subsections, in order to help the reader to follow each discussed point. Unfortunately, we cannot summarize the introduction more than a limit, as according to IJMS guidelines , review manuscripts have to contain at least 4000 words in the main body.

Moreover, only minimal language editing is required. We thank the reviewer for this suggestion. Done

Round 2

Reviewer 2 Report

1.       The first two pages of the article contain a brief introduction to the ROS/RNS species. The authors must include a figure to demonstrate the various reactive species generated. The authors must also include a figure to illustrate the reactive species that are generated by endogenous and exogenous sources and the various antioxidant defense mechanisms that restore normal redox state in melanocytes and the interaction with immune cells.

Response: We thank the Reviewer for this suggestion. In the main text, a new figure has been added. Figure 1 describes the major reactive species sources in melanocytes. The increase of reactive oxygen species (H2O2, O2−•) and/or reactive nitrogenous species (NO and ONOO- ) induces severe damages to major biomolecules, resulting in DNA and protein oxidation, as well as lipoperoxidation, that compromise cellular structure and function. Consequently, these alterations can induce inflammation and can initiate tumorigenesis processes (e.g. cell proliferation and adaptive immune resistance). To maintain acceptable levels of reactive species, melanocytes cells usually increase their antioxidant systems to protect cells from oxidative stress damage and restore physiological redox balance. The redox balance in the cell is normally regulated by a complex antioxidant system. Endogenous antioxidants include catalase (CAT), superoxide dismutase (SOD), glutathione peroxidases (GPXs) and glutathione (GSH). In particular, GSH metabolism protects melanocytes from the toxic effects of H2O2 formed during melanin synthesis. GSH metabolism, therefore, appears to be critically important to the maintenance of melanocyte cell viability [1]. Instead, natural antioxidant compounds can be obtained from the diet, e.g. beta-carotene (vitamin A), alpha-ascorbic acid (vitamin C), tocopherol (vitamin E) [2].

Query: The authors have not provided any information relating to the cross-talk between melanocytes and immune cells.

2.       Although the article title involves oxidative stress and immune response in melanoma, there is very little information provided on the immune response end. How do ROS/RNS species regulate immune cells? How do these reactive species promote melanoma growth and inflammation that results in an immune-suppressive tumor microenvironment? Various studies should be highlighted. For example, studies by Greten FR, Grivennikov SI et al., should be included.

Response: We thank the reviewer for this suggestion. We added new information in the section titled, Oxidative Stress in Melanocytes. Moreover, we modified the title of the manuscript. The new title is -Oxidative Stress in Melanoma: Ion Channels as a Target of Therapy. In comparison with other solid tumors, reactive species levels are particularly elevated in melanomas. Two important tissue characteristics may explain this further increase: the natural exposure to UV radiation and the presence of melanin. UV irradiation, a major contributor to skin cancer, can directly damage DNA via forming a large amount of cyclobutane pyrimidine dimers (CPDs), pyrimidine, pyrimidone adducts; moreover, UV induces the skin to produce, through photosensitizer molecules, high levels of ROS (singlet oxygen and hydroxyl radicals) immediately after irradiation and of RNS (NO and possibly ONOO−) at later timepoints. Free radical enhancements can also originate from UV-dependent activation of ROS-producing enzymes, such as nitric oxide synthases (NOXs), arachidonic acid cyclooxygenases (COXs), and lipoxygenases (LOXs). In addition, in melanoma cells the major source of ROS/RNS has been also attributed to the dysfunction of mitochondrial respiratory chain enzymes [2]. These alterations can induce inflammation and can further initiate tumorigenesis [11]. Reactive species and damaged DNA can activate intracellular protein complexes such as inflammasomes [12,13]. In this context, both keratinocytes and melanocytes secrete cytokines with pro-inflammatory action, thus modulating innate and adaptive immune responses [14]. All the immune-related molecules, cytokines, chemokines and nonimmune molecules, such as growth factors have both paracrine and autocrine effects upon the microenvironment and design the local milieu that initiates and then regulates local inflammation or can lose control, consequently favouring the process of tumorigenesis. Inflammation has acute and chronic stages, but its link to tumorigenesis is carried out by chronic inflammation [15]. During inflammatory response, mast cells and monocytes/macrophages are recruited [16]. In particular, mast cells are the first to migrate to the site of proliferation; macrophages follow later in the response. Both are capable of producing reactive species as a cytotoxic mediator to kill cells [17]. Reactive oxygen species can react with the nucleic acids attacking the nitrogenous bases and the sugar phosphate backbone and can evoke single- and double-stranded DNA breaks [18]. While acute inflammation is regulated by T-helper (Th)1-polarized T lymphocytes attracted by innate immune cells, secreting mainly anti-tumour immune molecules such as interleukin (IL)-2 and interferon (IFN)- γ, chronic inflammation is controlled by regulatory T cells (Tregs), Th2 cells, that secrete pro‐tumorigenic factors (IL‐4, IL‐6, IL‐10, IL‐13 and transforming growth factor (TGF)‐β) [9]. In this regard, reactive species produced by melanoma cells and tumour-infiltrating leukocytes, including Tregs, can suppress the immune responses [18]. We also reported in the section, titled Immune Evasion in Melanoma and Potential Novel Options for treatment, what it has been described. Melanoma is also an immunologic malignancy [19]. These cancer cells are constantly adapting to the host defenses by manipulating intrinsic and extrinsic biological pathways [19]. In the event of the onset and development of melanoma, the immune system is exposed to numerous previously unseen anti- gens that are derived from genetic abnormalities. In this context, the immune system can recognize and eliminate some cancers at an early stage of their development. The adaptive immune system appears to be of fundamental importance in the antitumor response, which is triggered by activation of a wide range of diverse and highly specific receptors on T and B cells. An effective immune response begins when the T or B cell recognizes the tumour antigen in a pro-stimulatory context and undergoes activation and proliferation. B cells have as receptor a surface IgM immunoglobulin and are able to recognize soluble antigens, bind to them and differentiate into plasmacytes, which secrete large amounts of highly specific antibodies [20]. Unfortunately, melanoma cells may develop numerous immuno-evasive mechanisms that allow them to resist natural or therapy-induced immune attacks [21]. Through these mechanisms, tumour cells are capable of modulating themselves and their surroundings in order to promote their survival, growth, and invasion, even under a persistent immune pressure. Indeed, stressors present in the tumour microenvironment, such as chronic hypoxia, play crucial roles in promoting tumour cell plasticity and heterogeneity, which finally leads to the acquisition of immune tolerance and tumour progression [22,23]. The plasticity of melanoma cells leads to a phenomenon called immune escape, whereby cancer cells acquire a less immunogenic phenotype and the ability to suppress anti- tumour immune cells within the tumour microenvironment [15,24,25].

Query: While the authors have provided some relevant information on the immune system front, the fact that they modified the original title/aim of the proposed manuscript to address the question proposed earlier is difficult to comprehend.

Author Response

The authors are grateful to the Reviewer who carefully revised our manuscript. Her/His suggestions have helped us in improving the manuscript. The manuscript has been corrected according to the suggestions of the Reviewer and a point by point reply is here provided.
